# Probing the symmetry breaking of a light–matter system by an ancillary qubit

Shuai-Peng Wang [1,2], Alessandro Ridolfo[3], Tiefu Li [4,5] ✉,
Salvatore Savasta[6] ✉, Franco Nori [7,8,9], Y. Nakamura [9,10] & J. Q. You [2] ✉

Hybrid quantum systems in the ultrastrong, and even more in the deep-strong, coupling regimes can exhibit exotic physical phenomena and promise new applications in quantum technologies. In these nonperturbative regimes, a qubit–resonator system has an entangled quantum vacuum with a nonzero average photon number in the resonator, where the photons are virtual and cannot be directly detected. The vacuum field, however, is able to induce the symmetry breaking of a dispersively coupled probe qubit. We experimentally observe the parity symmetry breaking of an ancillary Xmon artificial atom induced by the field of a lumped-element superconducting resonator deep-strongly coupled with a flux qubit. This result opens a way to experimentally explore the novel quantum-vacuum effects emerging in the deep-strong coupling regime.

Superconducting quantum circuits based on Josephson junctions (JJs)[1–8] have developed rapidly in recent years and demonstrated a quantum advantage, over classical counterparts, in information processing[9,10]. Now they are considered to be one of the most promising experimentally-realizable systems for quantum computing[11–13]. Also, the experimental advancements in superconducting qubit–resonator systems have stimulated theoretical and experimental research on quantum optics in the microwave regime[14,15]. As a solid-state version of cavity quantum electrodynamics (QED)[16,17], circuit QED[18–20] has greater flexibility and tunability, and it can achieve ultrastrong and even deepstrong light–matter couplings to individual qubits[21–27], owing to the large dipole moment of the superconducting qubit (i.e., artificial atom) and the small mode volume of the resonator. When the qubit–resonator coupling approaches the nonperturbative ultrastrong regime, novel quantum-optics phenomena occur[28–33], including puzzling modifications of the quantum vacuum of the system[34–40].

In the nonperturbative ultrastrong-coupling regime, the qubit–resonator system can be described by a quantum Rabi model. It is particularly interesting to harness controllable physical parameters to tune the quantum vacuum of the system, since it becomes a novel entangled ground state $|G\rangle$ rather than the trivial product ground state of the Jaynes–Cummings model. In such an exotic quantum vacuum, while the average photon number in the resonator is nonzero, i.e., $\langle G|a^\dagger a|G\rangle \neq 0$, where $a^\dagger(a)$ is the creation (annihilation) operator of the resonator mode, the ground-state photons are actually virtual (tightly bound to the artificial atom[41]) and cannot be directly detected. Theoretically, it was proposed to employ non-adiabatic modulations[37], sudden turn-off of the qubit–resonator interaction[38], or a spontaneous decay mechanism of multi-level systems[39] to convert these virtual photons into real ones (similar to the dynamical Casimir effect[42,43]), so as to generate radiation out of the resonator. However, these are still experimentally challenging.

[1]Quantum Physics and Quantum Information Division, Beijing Computational Science Research Center, Beijing 100193, China. [2]Interdisciplinary Center of Quantum Information, State Key Laboratory of Extreme Photonics and Instrumentation, and Zhejiang Province Key Laboratory of Quantum Technology and Device, School of Physics, Zhejiang University, Hangzhou 310027, China. [3]Dipartimento di Fisica e Astronomia, Università di Catania, 95123 Catania, Italy. [4]School of Integrated Circuits, and Frontier Science Center for Quantum Information, Tsinghua University, Beijing 100084, China. [5]Beijing Academy of Quantum Information Sciences, Beijing 100193, China. [6]Dipartimento di Scienze Matematiche e Informatiche, Scienze Fisiche e Scienze della Terra, Università di Messina, I-98166 Messina, Italy. [7]Theoretical Quantum Physics Laboratory, Cluster for Pioneering Research, RIKEN, Wako Saitama 351-0198, Japan. [8]Physics Department, The University of Michigan, Ann Arbor, MI 48109-1040, USA. [9]RIKEN Center for Quantum Computing (RQC), Wako, Saitama 351-0198, Japan. [10]Department of Applied Physics, Graduate School of Engineering, The University of Tokyo, Bunkyo-ku, Tokyo 113-8656, Japan. ✉e-mail: litf@tsinghua.edu.cn; salvatore.savasta@unime.it; jqyou@zju.edu.cn

In the standard model of particle physics, the $W^\pm$ and $Z$ weak gauge bosons obtain mass via the Higgs mechanism, in which the electroweak gauge symmetry SU(2) × U(1) is broken due to the interaction with a symmetry-broken vacuum field (the Higgs field) displaying a nonzero vacuum expectation value. In our experiment, we observe the parity symmetry breaking of a probe superconducting circuit (Xmon) dispersively coupled to a qubit–resonator system in the deep-strong coupling regime. This effect, although rather different (in our case, the broken symmetry is discrete and it is not spontaneous), shares some interesting analogies with the Higgs mechanism. At the optimal point, both the flux qubit and the qubit–resonator system have a well-defined parity symmetry[44]. In this parity-symmetry case, the quantum-vacuum expectation value of the resonator field is zero, $\langle G|(a + a^\dagger)|G\rangle = 0$, where $|G\rangle$ is the qubit–resonator ground state. With the external flux tuned away from the optimal point, parity-symmetry breaking is induced in the flux qubit and, in the presence of a very strong qubit–resonator coupling, it also significantly affects the resonator vacuum, giving rise to $\langle G|(a + a^\dagger)|G\rangle \neq 0$[35]. In our experiment, the achieved qubit–resonator system is in the deep-strong coupling regime, so the quantum-vacuum state is very different and, when away from the optimal point, this can produce a sizable nonzero value of $\langle G|(a + a^\dagger)|G\rangle$ as well as observable symmetry breaking effects. Indeed, as demonstrated in our experiment, the qubit–resonator system is able to break the parity selection rule of the Xmon dispersively coupled to the resonator, thus enabling forbidden transitions. We should emphasize that the similarity between the Higgs mechanism and our observation only comes from two key features: (i) the symmetry-broken vacuum has a nonzero expectation value and (ii) the field with a nonzero expectation value can induce symmetry breaking in another quantum system, in the absence of real excitations of the field. Of course, a system composed of just two qubits and a lumped-element resonator cannot fully reproduce the far more complex Higgs model.

## Results

### Deep-strongly coupled qubit–resonator circuit

The system is composed of a five-junction flux qubit deep-strongly coupled to a superconducting lumped-element resonator via a common Josephson junction (JJ) (Fig. 1). In addition, we use an Xmon as a quantum detector, which is capacitively coupled to the lumped-element resonator on the left and to a coplanar-waveguide resonator on the right. The whole device is placed in a dilution refrigerator cooled down to a temperature of ~30 mK.

Similar to the three-junction flux qubit[2], the five-junction flux qubit has both clockwise and counterclockwise persistent-current states. Away from the optimal point $\Phi_{ext} = (n + \frac{1}{2})\Phi_0$, where $\Phi_{ext}$ is the external flux threading the loop of the flux qubit, $\Phi_0 = h/2e$ is the superconducting flux quantum, and $n$ is an integer, these two persistent-current states have an energy difference $\varepsilon = 2I_p\delta\Phi_{ext}$, depending on the maximum persistent current $I_p$ and the flux bias $\delta\Phi_{ext} \equiv \Phi_{ext} - (n + \frac{1}{2})\Phi_0$. Also, there is a barrier between these two persistent-current states, which removes their degeneracy at the optimal point by opening an energy gap $\Delta$. In the basis of eigenstates, the Hamiltonian of the flux qubit can be written as (setting $\hbar = 1$) $H_q = \omega_q\sigma_z/2$, where $\omega_q = \sqrt{\Delta^2 + \varepsilon^2}$ is the transition frequency of the qubit and $\sigma_z$ is a Pauli operator. The quantum two-level system is a good model for the flux qubit because of its relatively large anharmonicity.

Compared to the coplanar-waveguide resonator, the lumped-element resonator has the advantage of only a single resonator mode[24]: $H_r = \omega_r a^\dagger a$, where $\omega_r$ is the resonance frequency of the resonator mode. This $\omega_r$ is V-shaped versus $\delta\Phi_{ext}$ around the optimal point[24] because the inductance across the qubit loop, as part of the

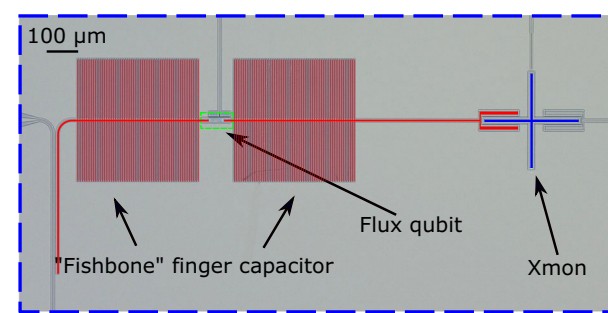

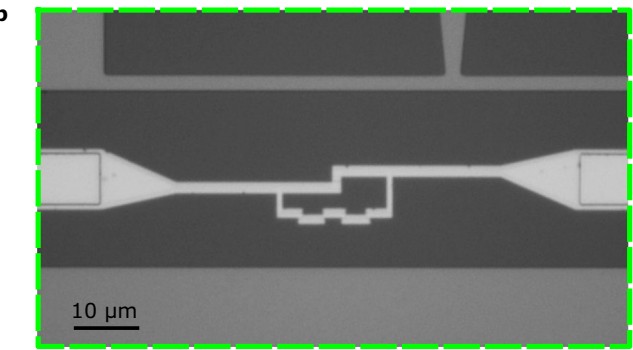

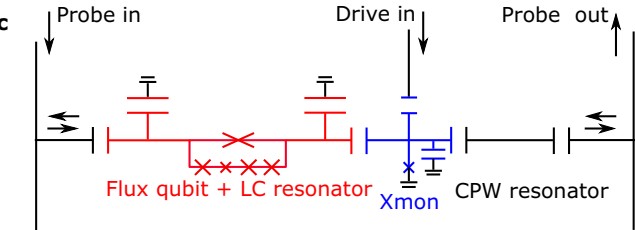

**Fig. 1 | Device. a** Optical image of the device. The lumped-element resonator is composed of two identical large "fishbone" interdigitated capacitors and a center conductor in between. The flux qubit consists of three identical larger JJs and a smaller JJ reduced by a factor of 0.42 in area. To enhance the coupling between the flux qubit and the lumped-element resonator, an even larger JJ (with its area doubled) is added to the qubit loop and shared with the center conductor. Extending to the left (right) is another section of 50 Ω coplanar-waveguide which couples to the input signal line (the Xmon qubit). **b** Zoom-in optical image of the area denoted by the green rectangular box in **a**. **c** Circuit diagram of the device (cf. Supplementary Fig. 1).

total inductance of the lumped-element resonator, depends approximately linearly on $|\delta\Phi_{ext}|$. The large JJ shared by the flux qubit and the lumped-element resonator acts as an effective inductance to produce an interaction between them, $H_{int} = g[\cos\theta\,\sigma_z - \sin\theta\,\sigma_x](a^\dagger + a)$, where $\tan\theta = \Delta/\varepsilon$, and $g = MI_pI_r$ is the coupling strength, with $M \approx L_c$ being the mutual inductance and $I_r = \sqrt{\omega_r/2(L_0 + L_c)}$ the vacuum fluctuation current along the center conductor of the lumped-element resonator, where $L_c$ is the inductance of the large JJ and $L_0$ is the geometry inductance. When the qubit–resonator coupling is in the ultrastrong or deep-strong regime, one cannot apply the rotating-wave approximation (RWA) to $H_{int}$, and the Hamiltonian of the qubit–resonator system is written as

$$H_s = \frac{1}{2}\omega_q\sigma_z + \omega_r a^\dagger a + g[\cos\theta\,\sigma_z - \sin\theta\,\sigma_x](a^\dagger + a)\,, \quad (1)$$

i.e., the generalized quantum Rabi model[15].

We can extract the parameters in $H_s$ by fitting the reflection spectra of the qubit–resonator system, as measured by applying a

probe tone to the system. Around $\omega_p/2\pi = 4.8$ GHz (near the bare frequency of the lumped-element resonator) and 5.6 GHz, clear transitions are observed; of which the corresponding frequencies are found to be consistent with the transition frequencies from the ground state $|G\rangle \equiv |0\rangle_s$ to the first- and second-excited states, $|1\rangle_s$ and $|2\rangle_s$ of the qubit–resonator system, respectively, i.e., $\omega_{01}$ and $\omega_{02}$ (see the solid fitting curves in Fig. 2c, b). Around $\omega_p/2\pi = 11.9$ GHz, we observe the transition from the ground state $|G\rangle$ to the third-excited state $|3\rangle_s$ (Fig. 2a), with the solid fitting curves corresponding to $\omega_{03}$. Moreover, similar to those in ref. 26, additional transitions are observed in Fig. 2a, which are attributed to the sideband transitions (the dashed curves in Fig. 2a) involving the Xmon levels as well, see Supplementary Information.

Near 5.6 and 11.9 GHz, the transmission background of the probe tone changes abruptly, forming two band edges (see Supplementary Fig. 2). The qubit–resonator system coupled to the band edges in Fig. 2a and b is analogous to the case of an atom coupled to a band edge in a photonic crystal waveguide[45,46]. The abrupt changes in the transmission background originate from the wire bonding and filters. Near the band edge, a photon emitted by the atom (in our case, it is the qubit–resonator system) is Bragg reflected and reabsorbed, resulting in the emergence of spectrally resolvable polariton states (similar to the vacuum Rabi splitting), which will disappear away from the band edge[45].

By fitting the transition frequencies $\omega_{01}$, $\omega_{02}$, and $\omega_{03}$ with experimental results in Fig. 2, we can derive the parameters of the generalized Rabi model in Eq. (1), which are $I_p = 245$ nA and $\Delta/2\pi = 15.0$ GHz for the flux qubit, $\omega_r/2\pi = 4.82$ GHz for the lumped-

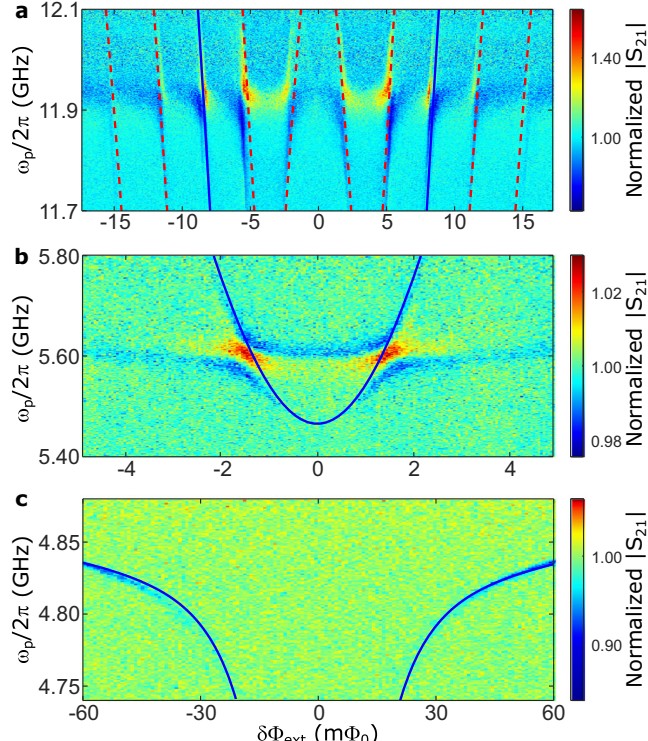

**Fig. 2 | Reflection spectra.** Reflection spectra of the deep-strongly coupled qubit–resonator system versus the external flux bias $\delta\Phi_{ext}$ and the probe frequency $\omega_p$ around $\Phi_{ext} = (3 + \frac{1}{2})\Phi_0$ (which is a more stable flux bias point than $\Phi_{ext} = \frac{1}{2}\Phi_0$ in our system). The solid blue curves in **a**–**c** are the fitted transition frequencies between the ground state to the third-, second- and first-excited states of the qubit–resonator system, respectively (i.e., $\omega_{03}$, $\omega_{02}$, and $\omega_{01}$). In **a**, the additional transitions indicated by the dashed red curves correspond to sideband transitions (assisted by the Xmon levels) in the system. Source data are provided as a Source Data file.

element resonator, and $g/2\pi = 4.55$ GHz for the qubit–resonator coupling. Here the obtained resonance frequency $\omega_r/2\pi = 4.82$ GHz is the value when the external flux bias is at the optimal point $\delta\Phi_{ext} = 0$. In our qubit–resonator system, we achieve $g/\omega_r \approx 0.944$, indicating that it indeed reaches the deep-strong coupling regime $g/\omega_r \sim 1$.

When $\delta\Phi_{ext} = 0$ ($\theta = \pi/2$), the deep-strongly coupled system in Eq. (1) reduces to the standard quantum Rabi model. Instead of a trivial (product) ground state $|g, 0\rangle$ in the Jaynes–Cummings model, it has a quantum vacuum (i.e., the entangled ground state) $|G\rangle$, with $\langle G|a^\dagger a|G\rangle \neq 0$. This standard Rabi model has a well-defined parity symmetry, characterized by $\sigma_z e^{i\pi a^\dagger a}$, which ensures that the ground state is a superposition of all states with an even number of excitations[28]. For this quantum vacuum, $\langle G|(a + a^\dagger)|G\rangle = 0$. When $\delta\Phi_{ext} \neq 0$, $H_s$ in Eq. (1) has an extra longitudinal coupling term proportional to $\sigma_z$. It breaks the parity symmetry of the model, and hence both even and odd numbers of excitations are allowed in the new ground state[35] as well as in the excited states of the coupled system. Now, both $\langle G|a^\dagger a|G\rangle \neq 0$ and $\langle G|(a + a^\dagger)|G\rangle \neq 0$.

## Detection of the induced symmetry breaking

Below we harness an Xmon[7] to detect the symmetry breaking of the lumped-element resonator. The Xmon is both largely detuned and weakly coupled to it via a small capacitor (cf. Fig. 1a). In such a dispersive regime, the effect of the Xmon on the qubit–resonator system is greatly reduced. The Xmon can be modeled by the Hamiltonian $H_X = 4E_c n^2 - E_J \cos\varphi$, where $E_c$ is the single-electron charging energy of the JJ, $E_J$ is the Josephson coupling energy, $n = -i\partial/\partial\varphi$, and $\varphi$ is the phase drop across the JJ. In the Xmon, the metallic cross and the ground metal provide the JJ with a large shunt capacitor to reduce its sensitivity to the charge noise[5,6].

The Xmon's parameters can be determined with the dispersive readout technique by coupling the Xmon to a coplanar-waveguide resonator (see Fig. 1c and Supplementary Fig. 1). The resonance frequency of this waveguide resonator is measured to be $\omega_{CPW}/2\pi = 3.554$ GHz and the coupling strength between the waveguide resonator and the Xmon qubit is $g_X/2\pi = 28$ MHz. Then, we obtain the transition frequency $\omega_X/2\pi = 5.181$ GHz of the Xmon qubit and its anharmonicity $A/2\pi = -0.16$ GHz. With these parameters as well as the relations $\omega_X = \sqrt{8E_cE_J} - E_c$ and $A = -E_c$, we have $E_c/2\pi = 0.16$ GHz and $E_J/2\pi = 20.97$ GHz. Because the lumped-element resonator couples to the Xmon, it induces an offset charge to the Josephson junction, leading $H_X \rightarrow \tilde{H}_X = 4E_c(n - n_R)^2 - E_J \cos\varphi$, where $n_R = i\frac{g'}{8E_c}(a - a^\dagger)$, and $g' \approx g_X$ (by a symmetric design) is the coupling strength between the lumped-element resonator and the Xmon qubit.

The total Hamiltonian of the deep-strongly coupled qubit–resonator system plus the Xmon can be expressed as $H_{tot} = H_s + \tilde{H}_X$. Owing to the large transition frequency between the ground and first-excited states, the deep-strongly coupled qubit–resonator system nearly stays in the ground state $|G\rangle$ at the temperature of ~30 mK.

Note that in the dispersive regime, where the Xmon–resonator coupling rate is much lower than the corresponding detuning, the energy transitions of the Xmon are almost unaffected by the interaction (i.e., flux bias-insensitive) and can be easily identified with standard spectroscopic techniques. We also observe that, neglecting the interaction between the resonator and the flux qubit, or considering a flux qubit at the optimal point, the Xmon–resonator system displays parity symmetry. On the contrary, when the qubit is brought out of the optimal point, the very strong qubit–resonator coupling strength can induce a symmetry breaking of the Xmon, even for moderate Xmon–resonator coupling strengths (see Supplementary Information).

In Fig. 3a, b, we show the single- and two-photon transitions between the lowest two levels of the Xmon using two-tone spectroscopy. Here the resonance frequency of the single-photon transition

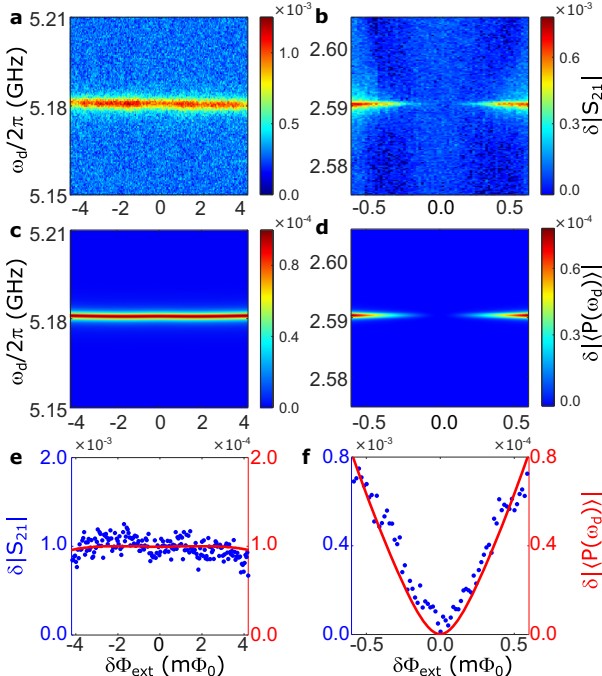

**Fig. 3 | Excitation spectra.** Excitation spectra of the Xmon qubit versus the external flux bias $\delta\Phi_{ext}$ and the drive frequency $\omega_d$ around $\Phi_{ext} = (3 + \frac{1}{2})\Phi_0$. The frequency of the probe tone is fixed at 3.554 GHz, in resonance with the $\lambda/2$ mode of the coplanar-waveguide resonator. **a** and **b** show the experimental results, corresponding to the single- and two-photon transitions of the Xmon qubit with frequencies $\omega_X$ and $\frac{1}{2}\omega_X$, respectively. **c** and **d** show the simulated results. The theoretical calculations display the changes in the amplitude of the Xmon polarization $|\langle P(\omega_d)\rangle|$. Loss rates for the flux qubit, lumped-element resonator, and Xmon are chosen to be $\gamma^{(q)}/2\pi = \gamma^{(a)}/2\pi = \gamma^{(b)}/2\pi = 2$ MHz, for simplicity and to be consistent with the observed linewidth in **a**. **e**, **f** Cross sections along the excitation spectra in **a** (**b**) and **c** (**d**) when $\omega_d/2\pi = 5.181$ GHz (2.5905 GHz). Source data are provided as a Source Data file.

corresponds to the transition frequency $\omega_X$ of the Xmon qubit, and the resonance frequency of the two-photon transition is $\frac{1}{2}\omega_X$. In our chip, $\frac{1}{2}\omega_X$ is designed to be well separated from both $\omega_{01}$ and $\frac{1}{2}\omega_{02}$ of the deep-strongly coupled qubit–resonator system to avoid any unwanted transitions. The drive power (−65 dBm) applied at the local drive port for exciting the two-photon transition is much stronger than that for the single-photon transition (−120 dBm). In Fig. 3b, the signal of the two-photon transition is found to disappear at the optimal point $\delta\Phi_{ext} = 0$, evidencing that the well-defined parity symmetry of the standard Rabi model preserves the parity selection rule of the Xmon. When deviating from the optimal point, the parity-symmetry breaking in $H_s$, in addition to producing a nonzero vacuum expectation value $v = \langle G|(a + a^\dagger)|G\rangle \neq 0$, is able to break the parity symmetry of the Xmon artificial atom, without however inducing any $\delta\Phi_{ext}$ dependent Lamb shift. Figure 3b shows that even a very small deviation from the parity symmetry point of the system Hamiltonian, which can be quantified by the adimensional parameter $\cot\theta = \varepsilon/\Delta \simeq 10^{-2}$, is able to activate two-photon transitions in the Xmon, in agreement with the theoretical calculations in Fig. 3d.

These results can be described by adopting a simplified model for the Xmon using only its four lowest energy levels. Considering the large detunings ($\gtrsim 20$ GHz), the effects from higher levels are negligible. With the Xmon now approximated as a four-level system (qudit), the total Hamiltonian of the qubit–resonator system plus the Xmon can be written as

$$H_{tot} = H_s + H_X^{(4)} - g'(a - a^\dagger)(b - b^\dagger), \qquad (2)$$

where $b = \sum_{n=0}^{3} \sqrt{n+1}|n\rangle\langle n+1|$ is the annihilation operator for the Xmon, and $H_X^{(4)} = \sum_{n=0}^{3} \varepsilon_n|n\rangle\langle n|$ is the bare Xmon energy ($H_X$), projected into the reduced four-dimensional Hilbert space. We can evaluate the single- and two-photon absorptions under the coherent drive of the Xmon by studying its effective polarization $\langle P(\omega_d)\rangle = \text{Tr}[-i(b - b^\dagger)\rho(\omega_d)]$, with $\rho$ being the density operator of the system. The latter can be calculated using the master equation approach in the dressed picture[47] (see Supplementary Information). The simulated results are shown in Fig. 3c, d, which are in good agreement with the experimental observations. This demonstrates the symmetry breaking of a quantum system which is coupled to the vacuum of another quantum system displaying symmetry breaking. Note that the deep-strongly coupled qubit–resonator system nearly stays in the ground state $|G\rangle$ at a temperature of ~30 mK, and in the present case of dispersive coupling with a largely detuned Xmon. Moreover, no real excitations of this system are coherently generated by the coherent drive at $\omega_d \simeq \omega_X/2$. We also point out that here the symmetry breaking is not spontaneous but due to the parity symmetry breaking of $H_s$ in Eq. (2), induced by the presence of a flux offset applied to the flux qubit. However, the adimensional parameter $\varepsilon/\Delta$, quantifying the degree of symmetry breaking induced by the flux offset on the flux qubit, is very small ($\varepsilon/\Delta \simeq 10^{-2}$) at $\delta\Phi_{ext} = 0.1$ m$\Phi_0$, when the Xmon two-photon transitions start to be observed (see Fig. 3b), and it does not affect the transition frequency of the Xmon. According to the additional calculations shown in Supplementary Fig. 6, the two-photon signals of the Xmon disappear if the effective coupling $g/\omega_r$ between the flux qubit and the LC resonator is reduced to 0.6. This provides evidence that the observed induced symmetry breaking of the Xmon is a unique feature in the near deep-strong coupling regime.

We observe that the interaction-induced symmetry-breaking mechanism detected here is more complex with respect to the Higgs mechanism and to that described in ref. 35. In these two cases, the effect is directly induced by the vacuum expectation value of the field. For example, for $w = \langle G|(a - a^\dagger)|G\rangle \neq 0$, the Xmon–resonator interaction in Eq. (2) could be approximated as $\sim -g'w(b - b^\dagger)$. It can be shown that this term directly determines the symmetry breaking of the probe qubit in ref. 35. However, in the present case, it turns out that $w = 0$, since the inductive coupling between the resonator and the flux qubit determines $w = 0$ and $v \neq 0$. Nonetheless, a full quantum analysis (see Supplementary Information) shows that in such a case ($w = 0$) as well, the Xmon can undergo symmetry breaking, when interacting with a field with no real excitations and displaying symmetry breaking. Using a probe qubit which is inductively coupled to the resonator would give rise to a symmetry-breaking mechanism directly determined by the nonzero vacuum expectation value $v = \langle G|(a + a^\dagger)|G\rangle \neq 0$. In the present case, the symmetry breaking of the qubit–resonator system determines a nonzero matrix element entering the Xmon two-photon transition rate; thus enabling two-photon transitions in the Xmon. Considering the eigenstates of the total Hamiltonian $H_{tot}$, the two-photon transition rate is proportional to the product $|Y_{0,1}Y_{1,2}|$, where $Y_{i,j} = \langle E_i| - i(b - b^\dagger)|E_j\rangle$, with $|E_j\rangle$ eigenvectors of $H_{tot}$ sorted from the lower to the higher corresponding energy levels. Thus, with $|E_0\rangle$ being the ground state of the whole interacting system, we identify $|E_1\rangle$ as the first-excited level of $H_s$ (slightly dressed by the interaction with the Xmon) and $|E_2\rangle$ the corresponding first excited dressed level of $H_X^{(4)}$. It turns out that $Y_{0,1}$ as well as $Y_{0,2}$ are nonzero and almost constant in the interval of flux offset reported here, while $Y_{1,2}$ is very well approximated by a linear function of $\delta\Phi_{ext}$ (see Supplementary Fig. 4) and it is zero for $\delta\Phi_{ext} = 0$, due to the parity symmetry. This explains the onset of the parity-symmetry breaking felt by the Xmon.

## Discussion

One interesting future possibility is that the current experimental method could be used to characterize the spontaneous vacuum symmetry breaking in the Dicke model (i.e., equilibrium superradiant phase

transition) when more flux qubits are integrated into the lumped-element resonator and operated at the optimal point simultaneously. If an equilibrium superradiant phase transition occurs, the small gap in the two-photon spectra of the Xmon in Fig. 3b will disappear. This means that two-photon transitions could be observed even for $\varepsilon/\Delta \simeq 0$. Note that in the present case, with a resonator interacting very strongly with only one flux qubit, we observe these parity-forbidden transitions for values $\varepsilon/\Delta \ll 1$ (specifically $\varepsilon/\Delta \gtrsim 0.01$). We also point out that, considering a setup with the capacitively coupled Xmon replaced by a galvanically coupled artificial atom (e.g., a flux qubit), the measured rate of parity-forbidden one- or two-photon transitions, would provide a direct measurement of the vacuum field expectation value $\langle G|(a+a^{\dagger})|G\rangle$, with a rate proportional to its square modulus[35].

In conclusion, we have experimentally probed the symmetry breaking of a lumped-element resonator, by observing the activation of two-photon transitions in a probe artificial atom (Xmon). The latter is dispersively coupled to the resonator and probed in the absence of any real coherent excitation of the resonator field; which however, displays a nonzero vacuum expectation value, as confirmed by theoretical calculations. The violation of the Xmon parity selection rule comes from virtual paths enabled by its interaction with an electromagnetic resonator whose parity symmetry is significantly broken by the deep-strong light–matter interaction with a flux qubit. The experimental results are in very good agreement with our theoretical analysis. The proposed setting offers a novel way to explore quantum-vacuum effects emerging in the light–matter ultrastrong and deep-strong coupling regimes and can be used as a tool to explore the coherence properties of quantum vacua in these exotic hybrid quantum systems[48–50], and the occurrence of superradiant phase transitions in Dicke-like systems[36].

## Methods

The experimental setup is shown in Supplementary Fig. 1. The superconducting lumped-element and coplanar-waveguide resonators are fabricated by patterning a niobium thin film of thickness 50 nm deposited on a $10 \times 3$ mm$^2$ silicon chip via electron beam lithography. The flux qubit is also fabricated on the silicon substrate in the middle of the center conductor of the lumped-element resonator by using both electron beam lithography and double-angle evaporation of aluminum. An external magnetic field generated by a magnetic coil surrounding the device is applied to tune the magnetic flux threading through the qubit loop. The Josephson junction in the Xmon is connected to the cross-shaped capacitor at one end and fabricated using separate steps of electron beam lithography and double-angle evaporation. Reflection spectra of the deep-strongly coupled qubit–resonator system at the frequency $\omega_p$ of the probe tone are measured with a vector network analyser (VNA). Another microwave signal at frequency $\omega_d$ is further applied at the local drive port of the Xmon for two-tone spectroscopy measurements. The input signals are attenuated and filtered at various temperature stages before finally reaching the sample. Also, two isolators and a low-pass filter (LPF) are used to protect the sample from the amplifier's noise.

## Data availability

The data that support the findings of this study are available from the corresponding authors upon reasonable request. Source data are provided with this paper.

## Code availability

The code that supports the findings of this study are available from the corresponding authors upon reasonable request.

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

## Acknowledgements

This work is supported by the National Key Research and Development Program of China (Grant No. 2022YFA1405200), the National Natural Science Foundation of China (Grants No. 92265202, No. 11934010, No. 62074091, and No. U2230402), Tsinghua University Initiative Scientific Research Program. A.R. acknowledges the QuantERA grant SiUCs (Grant No. 731473), the PNRR MUR project PE0000023-NQSTI, and the ICSC C Centro Nazionale di Ricerca in High-Performance Computing, Big Data and Quantum Computing. S.S. acknowledges the Army Research Office (ARO) (Grant No. W911NF1910065). F.N. is supported in part by Nippon Telegraph and Telephone Corporation (NTT) Research, the Japan Science and Technology Agency (JST) [via the Quantum Leap Flagship Program (Q-LEAP), and the Moonshot R&D Grant No. JPMJMS2061], the Asian Office of Aerospace Research and Development (AOARD) (Grant No. FA2386-20-1-4069), and the Foundational Questions Institute Fund (FQXi) (Grant No. FQXi-IAF19-06). Y.N. was partly supported by the Japan Society for the Promotion of Science (JSPS) Grants-in-Aid for Scientific Research (KAKENHI) (Grant No. 22H04937).

## Author contributions

S.-P.W., T.L., and J.Q.Y. conceived the experiment. S.-P.W. performed the experiment. S.-P.W. and A.R. analysed the data. A.R. and S.S. developed the theory. S.-P.W., A.R., T.L., S.S., F.N., Y.N., and J.Q.Y. discussed the results and contributed to the writing of the manuscript.

## Competing interests

The authors declare no competing interests.
