## [Peer Review File · Nature Communications]

REVIEWER COMMENTS

Reviewer #1 (Remarks to the Author):

In my previous report, I already pointed out that in this work, the authors demonstrate a very interesting circuit QED device, which can be operated in the deep-strong coupling regime and probed using a second, weakly coupled transmon qubit. It is experimentally shown that this device can be used to detect symmetry-breaking effects in such ultrastrongly coupled cavity QED systems.

I think these results are a nice proof-of-concept demonstration that will inspire further experimental and theoretical work in this direction. Compared to the originally submitted version of the manuscript, the authors have de-emphasized the relation to the Higgs symmetry breaking mechanism and also stated clearly that the symmetry breaking that they observe is induced and not spontaneous. I think this revised presentation is much precise and avoids any misleading claims. Therefore, I have no further objections and recommend a publication of this manuscript in *Nature Communications*.

Reviewer #3 (Remarks to the Author):

The authors report an experiment on a superconducting circuits consists of three parts: a lumped LC oscillator that is coupled to a flux qubit and Xmon qubit. At an optimal flux bias, the Hamiltonian for the entire system (Eq. 2) preserves the parity symmetry, namely, the even/odd parity of the total number of excitations is a good quantum number. In this case, the two-photon transition of a Xmon is not allowed due to the parity symmetry. When one moves away from the optimal point, the total system no longer conserves the parity of the total number of excitations and therefore in this case the two-photon transition of a Xmon is allowed. The authors present the experimental observation of this phenomena. The reported experimental results seem to agree well with the theory and that indicates the authors realized and measured a well-controlled circuits realizing a deep strong coupling limit.

The authors try to make a connection between the non-zero ground-state expectation value of $(a+a^\dagger)$ due to the non-zero flux bias and the change of selection rules for a two-photon transition due to the non-zero flux bias. They suggest the former induces the latter. In the abstract, for example, I quote, "We experimentally observe the parity symmetry breaking of an ancillary Xmon artificial atom induced by the field of a lumped-element superconducting resonator deep-strongly coupled with a flux qubit." I find this claim, which is one of the main points they make in the manuscript, rather misleading and unfounded. Both the non-zero ground-state expectation value of $(a+a^\dagger)$ and the change of selection rules are the consequences of reducing the symmetry of the total Hamiltonian (LC+flux qubit+Xmon) with a finite flux bias. In fact, as the authors' analysis shows, the rate for the two-photon transition is determined by the transition amplitude of $a+a^\dagger$ between two different energy eigenstates, not its ground-state expectation values.

Measuring the ground-state expectation value of a field operator would be indeed an important experiment. Contrary to the authors' claim, this aim is not achieved in the reported experiment. What they have observed is the quantities that become also non-zero, when one applies flux bias to make the ground-state expectation value of a field operator to be non-zero.

An superconducting circuit realizing the deep strong coupling limit with flux qubit and LC oscillators have already been achieved along with its spectroscopy, I don't find major innovations that may warrant a publication in Nature Communications in the current manuscript and therefore don't recommend its publication.

Reply to Reviewer #1's Comments

Comment:

In my previous report, I already pointed out that in this work, the authors demonstrate a very interesting circuit QED device, which can be operated in the deep-strong coupling regime and probed using a second, weakly coupled transmon qubit. It is experimentally shown that this device can be used to detect symmetry-breaking effects in such ultrastrongly coupled cavity QED systems.

I think these results are a nice proof-of-concept demonstration that will inspire further experimental and theoretical work in this direction. Compared to the originally submitted version of the manuscript, the authors have de-emphasized the relation to the Higgs symmetry breaking mechanism and also stated clearly that the symmetry breaking that they observe is induced and not spontaneous. I think this revised presentation is much precise and avoids any misleading claims. Therefore, I have no further objections and recommend a publication of this manuscript in Nature Communications.

Reply:

We thank the reviewer for the very positive comments and the recommendation for the publication.

Reply to Reviewer #3's Comments

Comment:

The authors report an experiment on a superconducting circuits consists of three parts: a lumped LC oscillator that is coupled to a flux qubit and Xmon qubit. At an optimal flux bias, the Hamiltonian for the entire system (Eq. 2) preserves the parity symmetry, namely, the even/odd parity of the total number of excitations is a good quantum number. In this case, the two-photon transition of a Xmon is not allowed due to the parity symmetry. When one moves away from the optimal point, the total system no longer conserves the parity of the total number of excitations and therefore in this case the two-photon transition of a Xmon is allowed. The authors present the experimental observation of this phenomena. The reported experimental results seem to agree well with the theory and that indicates the authors realized and measured a well-controlled circuits realizing a deep strong coupling limit.

The authors try to make a connection between the non-zero ground-state expectation value of $(a+a^\dagger)$ due to the non-zero flux bias and the change of selection rules

for a two-photon transition due to the non-zero flux bias. They suggest the former induces the latter. In the abstract, for example, I quote, “We experimentally observe the parity symmetry breaking of an ancillary Xmon artificial atom induced by the field of a lumped-element superconducting resonator deep-strongly coupled with a flux qubit.” I find this claim, which is one of the main point they make in the manuscript, rather misleading and unfounded. Both the non-zero ground-state expectation value of $(a+a^\dagger)$ and the change of selection rules are the consequences of reducing the symmetry of the total Hamiltonian (LC+flux qubit+Xmon) with a finite flux bias. In fact, as authors’s analysis shows, the rate for the two-photon transition is determined by the transition amplitude of $a+a^\dagger$ between two different energy eigenstates, not its ground-state expectation values.

Reply:

Reading this first part of the report, as well as other sentences below, our first impression is that this report has been written on the basis of the first version of our manuscript. Of course we suppose that this is not the case. However, in the second version, we have substantially revised the manuscript removing any claim potentially misleading or unfounded or that could be misinterpreted. We also added a number of new sentences in order to summarize the experimental findings in a more clear and transparent way. Compared to the first version of our manuscript (previously submitted to Nat. Phys.) we think that this second version is free from those issues, contrary to the claims of Reviewer #3.

Concerning the above paragraph in the Reviewer’s report, we fail to understand why the sentence in the abstract: “We experimentally observe the parity symmetry breaking of an ancillary Xmon artificial atom induced by the field of a lumped-element superconducting resonator deep-strongly coupled with a flux qubit.” would be misleading or unfounded. It is undeniable that the Xmon is directly (capacitively) coupled to the resonator field and that in the absence of this coupling the Xmon would display parity symmetry. Hence we find this criticism to be incorrect.

Comment:

Measuring the ground-state expectation value of a field operator would be indeed an important experiment. Contrary to the authors’ claim, this aim is not achieved in the reported experiment. What they have observed is the quantities that become also non-zero, when one apply flux bias to make the ground-state expectation value of a field operator to be non-zero.

Reply:

We agree with the Reviewer that a direct measurement of the ground-state expectation

value of a field operator would be indeed an important experiment. In our opinion it would be a very important experiment. Had this been the case we would have sent the work to Nature or Science journals. Unfortunately this is not the case. We have never stated this in our manuscript. However, our work provides a step towards this kind of measurements. For example, as shown in Ref. [2], if the probe qubit would be inductively coupled to the resonator, the observed parity symmetry breaking would provide a rather direct measurement of the non-zero ground-state expectation value of a field operator. In the revised manuscript, we have added two sentences to indicate that our work can provide this step, at the end of the first paragraph in the right column on page 5.

Comment:

An superconducting circuit realizing the deep strong coupling limit with flux qubit and LC oscillators have already been achieved along with its spectroscopy, I don't find major innovations that may warrant a publication in Nature Communications in the current manuscript and therefore don't recommend its publication.

Reply:

Although only very few experiments realizing the deep strong coupling limit with superconducting circuits have been reported, as can be inferred soon by reading our manuscript, this does not represent the main result of this work. Perhaps a brief summary of our results here can better show what this work adds:

1. The ultrastrongly coupled system that we consider is in the dispersive regime. Specifically the qubit transition frequency is more than three times higher of both the coupling rate and the resonance frequency of the resonator. Hence the lowest energy states of the system can be interpreted as dressed field states.
2. Thanks to the very strong coupling rate between the LC resonator and the flux qubit, a small flux offset applied to the qubit is able to affect the parity symmetry of the LC resonator, despite the two subsystems are largely detuned. We show theoretically that the huge coupling is able to induce a significant non-zero ground state expectation value in the system as well as to break the parity symmetry of the resonator excited energy states.
3. The latter effect in point 2 is experimentally demonstrated by observing the parity symmetry breaking of a probe qubit capacitively weakly coupled to the LC resonator.
4. The probe qubit (the Xmon) is weakly directly coupled to the LC resonator and the system is driven at a frequency which is far detuned from the resonator frequency, so that no real excitations are created in the ultrastrongly coupled system.

5. A theoretical analysis shows that the observed effect is enabled by the huge coupling between the resonator and the flux qubit. Reducing by small amount the coupling the effect is no more visible (Fig. S6).
6. As we wrote in the conclusions, these experimental results demonstrate a method which can be applied in the future to directly probe realizations of spontaneous symmetry breaking in systems implementing the Dicke model.

REVIEWERS' COMMENTS

Reviewer #3 (Remarks to the Author):

In the response, the authors have summarized the same claim presented in the manuscript. I will elaborate the way I view their experiment. The authors' experimental set up realizes the Hamiltonian given in Eq. (2). This is a composite system of coupled harmonic oscillator, flux qubit and transmon qubit. The transition amplitudes between energy levels of the transmon will therefore depend on the property of the energy eigenstates of the total Hamiltonian Eq. (2). Now, the parity symmetry of the total Hamiltonian Eq. (2) breaks down when the flux on the flux qubit is moved away from the symmetric point. Then, of course, some of the transition amplitudes between levels of transmon which were zero when the total parity symmetry is preserved will become nonzero and therefore certain transitions that are not allowed at the symmetry point becomes allowed away from the symmetry point. Their experiment is, in my view, therefore a spectroscopic evidence of the selection rules of the total Hamiltonian Eq. (2). The authors want to interpret this as the eigenstate properties of the Rabi Hamiltonian inducing a parity symmetry breaking of the qubit. I guess it is fine to have this view/interpretation, although I think the view point I summarized above is more straightforward. Now, a important question is by having this interpretation whether they add any new insight to the field. I think the answer to this question lies on whether their detection method will be indeed useful to detect the spontaneous symmetry breaking of the Rabi hamiltonian at the symmetric point. Without this connection, I fail to see any value in their interpretation of their experiment. In the current manuscript, neither in their reply, there is no convincing evidence that this will indeed be true. They just claim that this is a stepping stone to that goal without any support.

This is the basis of my recommendation and this has not changed after reading their response. Through a revision, If the authors can show that their detection method is indeed applicable to the measurement of spontaneous symmetry breaking and the change of vacua, then I would agree that breaking the symmetry by hand and see the signal is indeed an important stepping stone toward this goal that merits to be published here.

Reply to Reviewer #3's Comments

Comment:

In the response, the authors have summarized the same claim presented in the manuscript. I will elaborate the way I view their experiment. The authors' experimental set up realizes the Hamiltonian given in Eq. (2). This is a composite system of coupled harmonic oscillator, flux qubit and transmon qubit. The transition amplitudes between energy levels of the transmon will therefore depend on the property of the energy eigenstates of the total Hamiltonian Eq. (2). Now, the parity symmetry of the total Hamiltonian Eq. (2) breaks down when the flux on the flux qubit is moved away from the symmetric point. Then, of course, some of the transition amplitudes between levels of transmon which were zero when the total parity symmetry is preserved will become nonzero and therefore certain transitions that are not allowed at the symmetry point becomes allowed away from the symmetry point. Their experiment is, in my view, therefore a spectroscopic evidence of the selection rules of the total Hamiltonian Eq. (2). The authors want to interpret this as the eigenstate properties of the Rabi Hamiltonian inducing a parity symmetry breaking of the qubit. I guess it is fine to have this view/interpretation, although I think the view point I summarized above is more straightforward. Now, an important question is by having this interpretation whether they add any new insight to the field. I think the answer to this question lies on whether their detection method will be indeed useful to detect the spontaneous symmetry breaking of the Rabi Hamiltonian at the symmetric point. Without this connection, I fail to see any value in their interpretation of their experiment. In the current manuscript, neither in their reply, there is no convincing evidence that this will indeed be true. They just claim that this is a stepping stone to that goal without any support.

This is the basis of my recommendation and this has not changed after reading their response. Through a revision, If the authors can show that their detection method is indeed applicable to the measurement of spontaneous symmetry breaking and the change of vacua, then I would agree that breaking the symmetry by hand and see the signal is indeed an important stepping stone toward this goal that merits to be published here.

Reply:

There are two points raised by the reviewer, which we will reply one by one in the following:

1. The reviewer suggests another interpretation of our results, that the symmetry breaking of the probe Xmon is the result of the explicit symmetry breaking of the total Hamiltonian. We would say that actually it is not the case.

We observe that our manuscript already in the previous version included the interpretation suggested by Reviewer #3. In our manuscript we explicitly explained that the total system Hamiltonian is broken by the presence of a flux offset applied to the flux qubit. Please see, e.g., the main text in first paragraph right column on page 5:

“In the present case, the symmetry breaking of the qubit-resonator system determines a $\{\text{it nonzero}\}$ matrix element entering the Xmon two-photon transition rate, thus enabling two-photon transitions in the Xmon. Considering the eigenstates of the total Hamiltonian H_{tot} , the two-photon transition rate is proportional to the product ...”

See also the detailed analyses in the Supplementary Material. However, we agree that, at least in the main text the Reviewer’s interpretation can be made more explicit. We have now added (first paragraph left column on page 5) a sentence where it is clearly stated that the total Hamiltonian has a broken non-spontaneous symmetry:

“We point out that here the symmetry breaking is not spontaneous but due to the parity symmetry breaking of H_s in Eq.~\eqref{Hs}, induced by the presence of a flux offset applied to the flux qubit. However, the adimensional parameter ε/Δ , quantifying the degree of symmetry breaking induced by the flux offset on the flux qubit, is very small ($\varepsilon/\Delta \simeq 10^{-2}$) at $\delta \Phi_{\text{ext}} = 0.1 \text{ m}\Phi_0$ when the Xmon two-photon transitions start to be observed (see Fig. \ref{fig3}b) and does not affect the transition frequency of the Xmon. According to additional calculations shown in Fig. S6 (see Supplementary Information), the two-photon signals of the Xmon disappear if the effective coupling g/ω_r between the flux qubit and the LC resonator is reduce to 0.6. This evidences that the observed induced symmetry breaking of the Xmon is a unique feature in the near deep strong coupling regime.”

We also observe that the Reviewer’s interpretation is not in contrast with our analysis.

- a) First, the coupling between the probe Xmon and the LC resonator is dispersive and quite weak, as can be seen in Fig. 3 that the excitation spectra of the Xmon are flux-bias-insensitive. While parity symmetry breaking is common in flux-tunable qubits, it never happens in bias-independent qubits with fixed transition frequencies.
- b) Second, as has been shown in the Supplementary Fig.6, the two-photon signals of the Xmon will disappear if the effective coupling g/ω_r between the flux qubit and the LC resonator is reduced to 0.6. This evidences that the observed induced symmetry breaking of the Xmon is an unique feature in the near deep strong coupling regime. It will not happen when the Xmon is dispersively weakly coupled to a JC system ($g/\omega_r < 0.1$) or a single flux qubit, even though the latter is also explicitly symmetry-broken.

- c) Third, the combined effect of the deep strong coupling and the applied external flux bias is that both the system (flux qubit plus LC resonator) vacuum and the higher eigenstates are significantly changed. When the probe Xmon is dispersively weakly coupled to the system (especially the system is in its vacuum, i.e., no real excitations are generated), its parity symmetry is broken via its interaction with this special vacuum. The probe Xmon only couples with the system vacuum, and the coupling is dispersive and quite weak, thus a flux offset of the total system itself could not enable the symmetry breaking in the Xmon.
2. The reviewer concerns about the feasibility of our method to measure the spontaneous symmetry breaking at the optimal point. To further elucidate this point, we have added some new sentences in the main text.
- a) We have added the discussions about the adimensional parameter ε/Δ that quantifies the degree of symmetry breaking induced by the flux offset on the flux qubit. It can be seen (Fig. 3) that the adimensional parameter ε/Δ is very small ($\varepsilon/\Delta \simeq 0.01$ at $\delta\Phi_{\text{ext}} \simeq 0.1 \text{ m}\Phi_0$), when the Xmon two-photon transitions start to be observed. Notice that in the present case, with a resonator interacting very strongly with only one flux qubit, we observe these parity-forbidden transitions for values $\varepsilon/\Delta \ll 1$ (specifically $\varepsilon/\Delta \geq 0.01$). When more flux qubits are integrated in the LC resonator, the values of ε/Δ needed to observe the Xmon two-photon transitions are expected to be further reduced, and approach the optimal point ultimately if spontaneous symmetry breaking happens.
- b) We also point out that, considering a setup with the capacitively coupled Xmon replaced by a galvanically coupled artificial atom (e.g. a flux qubit), the measured rate of parity-forbidden one- or two-photon transitions, would provide a direct measurement of the vacuum field expectation value, being the rate proportional to its square modulus. A nonzero vacuum field expectation value is a direct signature of both the explicit (induced by the flux offset) and spontaneous symmetry breaking of the vacua of a light—matter coupled system, the measurement of which is a very valuable target for a future experimental study based on the current results.

We hope that the explanations above explain more clearly that our results are a symmetry breaking induced by the interaction with a quantum vacuum, totally different from the explicit symmetry breaking that is common in flux-tunable circuits.